# Hydrophobic Paper-Based SERS Sensor Using Gold Nanoparticles Arranged on Graphene Oxide Flakes

**DOI:** 10.3390/s19245471

**Published:** 2019-12-11

**Authors:** Dong-Jin Lee, Dae Yu Kim

**Affiliations:** 1Inha Research Institute for Aerospace Medicine, Inha University, Incheon 22212, Korea; voinaimir82@gmail.com; 2Department of Electrical Engineering, College of Engineering, Inha University, Incheon 22212, Korea

**Keywords:** hydrophobic paper, graphene oxide, surface-enhanced Raman scattering (SERS), gold nanoparticles arranged on graphene oxide flakes (AuNPs@GO)

## Abstract

Paper-based surface-enhanced Raman scattering (SERS) sensors have garnered much attention in the past decade owing to their ubiquity, ease of fabrication, and environmentally friendly substrate. The main drawbacks of a paper substrate for a SERS sensor are its high porosity, inherent hygroscopic nature, and hydrophilic surface property, which reduce the sensitivity and reproducibility of the SERS sensor. Here, we propose a simple, quick, convenient, and economical method for hydrophilic to hydrophobic surface modification of paper, while enhancing its mechanical and moisture-resistant properties. The hydrophobic paper (h-paper) was obtained by spin-coating diluted polydimethylsiloxane (PDMS) solution onto the filter paper, resulting in h-paper with an increased contact angle of up to ≈130°. To complete the h-paper-based SERS substrate, gold nanoparticles arranged on graphene oxide (AuNPs@GO) were synthesized using UV photoreduction, followed by drop-casting of AuNPs@GO solution on the h-paper substrate. The enhancement of the SERS signal was then assessed by attaching a rhodamine 6G (R6G) molecule as a Raman probe material to the h-paper-based SERS substrate. The limit of detection was 10 nM with an R^2^ of 0.966. The presented SERS sensor was also tested to detect a thiram at the micromolar level. We expect that our proposed AuNPs@GO/h-paper-based SERS sensor could be applied to point-of-care diagnostics applications in daily life and in spacecraft.

## 1. Introduction

Surface-enhanced Raman scattering (SERS) is a crucial tool for the analysis of molecule traces since the accidental discovery of the enhancement of Raman scattering signals upon Ag-roughened surfaces [1]. The dominant mechanism of SERS is characterized by locally enhanced electromagnetic (EM) fields occurring in the vicinity of the metal nanostructures owing to the localized surface plasmon resonances (LSPRs) [2,3,4]. The spectral positions of the LSPRs are functions of the dielectric constants of the surrounding media and structural information of the metal nanostructures, such as size, shape, and material [5]. According to theoretical calculations, the maximum enhancement factor (EF) using LSPR is around 10^11^ [6]. Another mechanism for SERS is chemical enhancement (EF of 10^1^–10^3^), which occurs via chemical interactions between the molecules and metal nanostructures [6]. Generally, the SERS enhancement is less than eight orders of magnitude [7].

Recently, flexible SERS sensors have garnered much research attention owing to their capability for in situ and onsite detection, thereby avoiding the complicated extractions of analytes and tedious sample preparation steps [4,8,9,10,11,12,13,14]. Among the available variety of flexible SERS substrates, paper substrates are promising candidates for cost-effective SERS sensors [15]. The main limitations of paper substrates for SERS sensors are their high porosity, inherent hygroscopic nature, and hydrophilic surface properties, which reduce the sensitivity and reproducibility of the sensors. Therefore, an appropriate surface modification process is required. Various surface modification techniques for paper substrates have been investigated in the past. Oh et al. developed a cellulose-nanofibril-coated paper substrate to reduce the pores and surface roughness [16]. Lee et al. proposed a filter paper functionalized with an alkyl ketene dimer for hydrophilic-to-hydrophobic surface property modification [17].

In this study, we propose a simple, convenient, timesaving, and economical method for hydrophobic surface modification of the filter paper and demonstrate the h-paper-based SERS sensor composed of gold nanoparticles arranged on graphene oxide (AuNPs@GO) flakes on the h-paper SERS substrate. The h-paper was prepared by spin-coating diluted polydimethylsiloxane (PDMS) on the filter paper, and the contact angle of the h-paper increased up to ≈130°. To fabricate a stable metal-nanoparticle platform, graphene oxide (GO) was used as a support for the arrangement of AuNPs. Graphene and graphene-based substrates are considered a promising material for SERS sensors owing to their large surface area and excellent molecule adsorption ability, and therefore, a number of graphene-based SERS devices have been proposed [7,18,19,20,21,22,23,24,25,26]. In addition, GO plays a crucial role as a fluorescence quencher and molecular stabilizer that render GO-metal NPs as hybrid platforms [27]. In this context, the AuNPs@GO was synthesized using UV photoreduction, resulting in approximately 6-nm AuNPs on the surfaces of graphene flakes. The paper-based SERS sensor was fabricated by placing a 50-μL drop of AuNPs@GO solution on the h-paper. The sensitivity was measured using rhodamine 6G (R6G) molecules with concentrations of 1 mM to 10 nM as the Raman probe material. The limit of detection (LOD) was 10 nM with an R^2^ of 0.966. We also investigated the presented SERS sensor to detect a thiram at the micromolar level.

## 2. Experimental Section

### 2.1. Materials and Reagents

Polydimethylsiloxane (PDMS, Sylgard 184) was purchased from Dow Corning, Midland, MI, USA. Heptane (for HPLC, 99%), gold (III) chloride solution (HAuCl_4_, 99.99%), rhodamine 6G (R6G), and thiram were purchased from Sigma-Aldrich Korea, Seoul, Korea. Graphene oxide solution (1 mg/mL, flake size of <1 μm) was purchased from Graphene Square, Suwon, Korea. Filter paper was purchased from Whatman, Piscataway, NJ, USA. All reagents were used without further purification.

### 2.2. Photosynthesis of AuNPs@GO

A photoreduction technique was used to arrange the AuNPs on the surfaces of the GO flakes using UV light irradiation, as reported in a previous work [27]. Briefly, 5 mM HAuCl_4_ solution (2 M methanol-deionized water solution) was prepared, and approximately 1 mg/mL of GO solution was dispersed in 20 mL of 5 mM HAuCl_4_ solution; the mixture was then sonicated for 10 min. A UV lamp (2 × 15 W, UVITEC, Cambridge, UK) with a central wavelength of 254 nm was then used to irradiate the above mixture for 60 min under magnetic stirring at 400 rpm. The suspension was subsequently centrifuged for 20 min at 10,000 rpm. The supernatant was discarded, and the sediments were redispersed in Milli-Q water. This process was repeated three times, and finally, 500 µL of the AuNPs@GO solution was prepared.

### 2.3. Fabrication of the h-Paper-Based SERS Sensor

PDMS with a 10:1 ratio of base to crosslinker by mass was prepared. The PDMS solution was diluted with heptane at different weight fractions (0 wt%, 40 wt%, 70 wt%, and 100 wt%), and the diluted PDMS solution was sonicated for 10 min to remove air bubbles. The h-paper was fabricated by spin-coating the diluted PDMS solutions on filter paper at 1000 rpm for 1 min. The PDMS was cured at room temperature for 48 h. The h-paper-based SERS substrate was completed by drop-casting 50 µL of the AuNPs@GO solution on the h-paper substrate. To evaluate the enhancement of the Raman scattering signal for the proposed SERS substrates, the AuNPs@GO/h-paper substrates were soaked in R6G in deionized (DI) water and thiram in ethanol of varying concentrations for 1 h. SERS measurements were performed after complete removal of the solvents by blowing nitrogen gas.

### 2.4. Characterization and Measurements

The morphological and chemical compositions of the h-paper-based SERS substrates were characterized using scanning electron microscopy (SEM; Hitachi S-4300SE, Hitachi, Tokyo, Japan), transmission electron microscopy/energy dispersive spectroscopy (TEM/EDS; Titan TM 80-300, Thermo Fisher Scientific, Waltham, MA, USA), X-ray photoelectron spectroscopy (XPS; K-Alpha, Thermo Fisher Scientific, Waltham, MA, USA), and UV-Vis spectroscopy (Lambda 750, Perkin-Elmer, Norwalk, Connecticut, CT, USA). Raman spectra were measured using Raman spectroscopy (LabRAm HR Evolution, HORIBA, Kyoto, Japan). Raman spectra of R6G molecules were obtained with a laser excitation wavelength of 532 nm, a laser power of 2 mW, an acquisition time of 10 s, and an accumulation of 5. For thiram molecules, Raman spectra were obtained in a laser excitation wavelength of 785 nm, a laser power of 2.4 mW, an acquisition time of 10 s, and an accumulation of 5. The contact angles were characterized using a homemade contact angle measurement system and ImageJ 1.52h version (NIH, Bethesda, MD, USA) [28].

## 3. Results and Discussion

The schematic representation of the fabrication process for the h-paper-based SERS substrate is presented in Figure 1. To arrange the AuNPs on the GO surfaces, GO solution (1 mg/mL in DI water) was dispersed in 20 mL of 5 mM HAuCl_4_ solution mixed with 2 M methanol-DI water under UV radiation for 60 min, as shown in Figure 1a. Methanol produces organic radicals, such as –CH_2_OH under UV light, thus accelerating the reduction of AuC_4_^−^ to Au^0^ on the GO surfaces. Figure 1b shows the fabrication process of the h-paper-based SERS substrate. After the hydrophobic treatment of the filter paper using the diluted PDMS solution, 50 µL of the AuNPs@GO solution was drop-cast on the h-paper. Because of hydrophobic modification of the filter paper, the absorption rate of the aqueous solution was lowered, providing a long retention time for the analyte solution. Furthermore, hydrophobic modification provides an enhancement of the mechanical endurance and reduction of the hygroscopic property [29]. Using R6G as the Raman probe, the enhancement of the SERS signal was investigated for the proposed h-paper-based SERS substrate.

Figure 2 shows the SEM images of the filter paper coated with diluted PDMS at 0, 40, 70, and 100 wt%. With the increase of the PDMS weight fraction, the fibrous nature of the filter paper gradually disappeared. It is worth noting that in our previous study, the PDMS-coated paper provided moisture-resistant properties [29]. The insets in Figure 2 show the optical microscopic images of water droplets for contact angle measurements. For the bare filter paper, the water droplet was momently absorbed into the paper, and the contact angle could not be measured. The contact angle of the h-paper with 40 wt% PDMS was about 130.6° ± 1.7°. The contact angle of the h-paper slightly decreased with a further increase of the PDMS weight fraction, as shown in Appendix A. The contact angle of the h-paper with 70 wt% and 100 wt% were about 128.4° ± 2.1° and 124.2° ± 3.3°, respectively. Even though the water contact angles slightly decreased, they still confirmed the hydrophobic surface property. The h-paper with 40 wt% PDMS was used as the hydrophobic substrate in subsequent studies owing to its adequate hydrophobic property.

To fabricate the h-paper-based SERS substrate, about 50 µL of the AuNPs@GO solution was drop-casted onto the h-paper substrate and dried at ambient conditions. Figure 3a,b shows the SEM images of the h-paper-based SERS substrate. The AuNPs@GO spot that formed was almost round with a smooth outline, and the spot area was about 4.2 mm^2^, as shown in Figure 3a. Figure 3b shows the interface between the h-paper and the AuNPs@GO/h-paper. Figure 3c shows the UV-Vis spectra of GO and AuNPs@GO dispersions. For the AuNPs@GO dispersion, a broad peak was observed at around at 564 nm, which was from the LSPRs of AuNPs. Figure 3d shows the XPS spectrum of the Au 4f peak recorded for AuNPs@GO. The Au 4f_7/2_ peak was observed at a binding energy of 82.9 eV, and the Au 4f_5/2_ peak was observed at 86.5 eV, indicating the binding of AuNPs to the surface of GO [30].

Figure 4 shows the morphological and chemical compositions of the AuNPs@GO. From the TEM images shown in Figure 4a,b, the AuNPs were dispersed well on the surfaces of the GO flakes. The average diameter and standard deviation of the AuNPs was about 7.3 nm and 1.9 nm, respectively, as shown in Figure 4c. The particle distribution of the AuNPs was obtained from eight regions of interest and 370 particles using ImageJ software, as shown in Appendix A [28]. Figure 4d shows the EDS spectrum of the AuNPs@GO. The Au characteristic peaks were also observed.

Figure 5 shows the SERS signal of the AuNPs@GO substrate with R6G molecules with concentrations in the range from 10^−3^ to 10^−8^ M. The R6G spectra at 10^−3^ M on the AuNPs@GO substrate exhibited strong peaks of the vibrational bands at about 612, 773, 1134, 1310, 1363, and 1651 cm^−1^, corresponding to the Raman characteristic peaks of R6G (Appendix A), as shown in Figure 5a. Figure 5b shows the SERS spectra of R6G molecules with concentrations in the range from 10^−3^ to 10^−8^ M. The most intense R6G Raman peak was around 612 cm^−1^, and this peak was selected to evaluate the SERS activity of the h-paper-based SERS substrate. Figure 5c shows the Raman intensity at 612 cm^−1^ as a function of the logarithmic concentration of R6G on the AuNPs@GO for evaluation of the linear relationship between Raman intensity and logarithmic concentration. According to the linear fitting line, the fitted equation was I = 1319.8 + 153.4logC with an R^2^ of 0.966. The LOD was 10 nM, which was comparable with other sensors based on a graphene composite [31]. To investigate the uniformity of the proposed AuNPs@GO/h-paper SERS sensor, the Raman intensities at 612 cm^−1^ were evaluated from 12 different spots. The relative standard deviation (RSD) was about 14.2%. To evaluate the reproducibility of the proposed AuNPs@GO/h-paper SERS substrates, we fabricated 15 AuNPs@GO/h-paper SERS substrates, as shown in Appendix A and acquired the Raman spectra of R6G with 10^−3^ M. Appendix A shows the intensities of Raman peaks at 612 cm^−1^, and the corresponding RSD was about 11.3%. For a reliability test, we compared the sensitivity of AuNPs@GO/h-paper substrates that were kept for 0 and 15 days, as shown in Appendix A. After 2 weeks of aging under ambient conditions (day 15), the Raman intensity at 612 cm^−1^ slightly decreased by about 5.4% compared to the initial state (day 0).

To demonstrate the possibility of practical utilization, we investigated the SERS activity of AuNPs@GO/h-paper substrates with thiram, which is a representative fungicide used for the protection of fruits and vegetables [10]. Figure 6a shows the prominent Raman peaks at 441, 554, 1132, 1370, and 1505 cm^−1^ for thiram at 10^−3^ M [17]. The strongest peak at 1370 cm^−1^, which was attributed to a CH_3_ deformation vibration and C–N stretching vibration, was selected to evaluate the sensitivity of AuNPs@GO/h-paper substrates. Figure 6b shows the SERS spectra of thiram molecules with concentrations in the range from 10^−3^ to 10^−6^ M. Figure 6c shows the Raman intensity at 1370 cm^−1^ as a function of the logarithmic concentration of thiram on the AuNPs@GO. The LOD for thiram was 1 μM.

## 4. Conclusions

In summary, we developed a simple, convenient, timesaving, and economical method for hydrophobic surface modification of the filter paper and demonstrated the AuNPs@GO/h-paper-based SERS sensor. Hydrophobic treatment increased the contact angle and decreased the contact area of the aqueous solution. These brought about an extended retention time of the AuNPs@GO solution in the reduced contact area, resulting in the concentrated AuNPs@GO on h-paper, enabling the creation of SERS hot-spots. The sensitivity was measured using R6G molecules with concentrations from 1 mM to 10 nM as Raman probe materials. The limit of detection (LOD) was 10 nM with an R^2^ of 0.966. To demonstrate the possibility of practical utilization, the presented SERS sensor was tested to detect a thiram at the micromolar level. We expect that our proposed AuNPs@GO/h-paper-based SERS sensor could be applied for point-of-care diagnostics applications in daily life and in spacecraft.

## Figures and Tables

**Figure 1 sensors-19-05471-f001:**
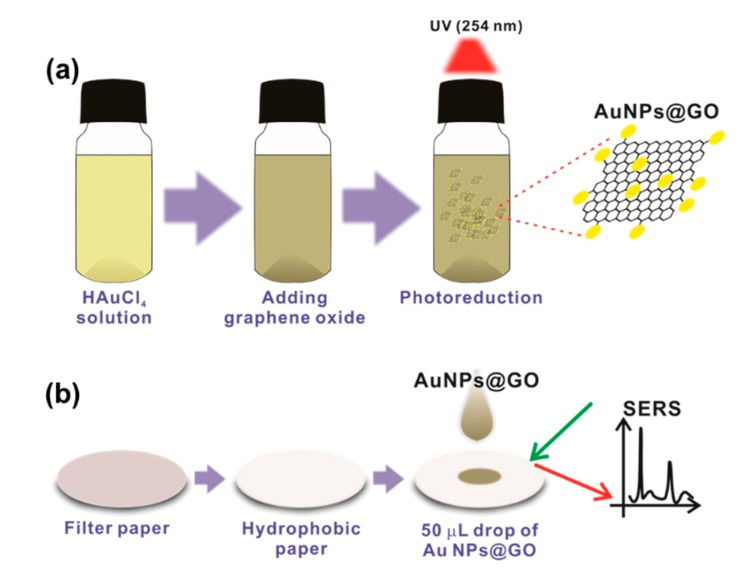
Schematic representation of the fabrication process for h-paper-based surface-enhanced Raman scattering (SERS) sensor. (**a**) Photoreduction process for the synthesis of AuNPs arranged on GO flakes (AuNPs@GO). Under UV irradiation, AuCl_4_^−^ ion was reduced to Au^0^ on the surface of the GO. (**b**) The h-paper-based SERS substrate was fabricated by drop-casting 50 µL AuNPs@GO solution onto the h-paper.

**Figure 2 sensors-19-05471-f002:**
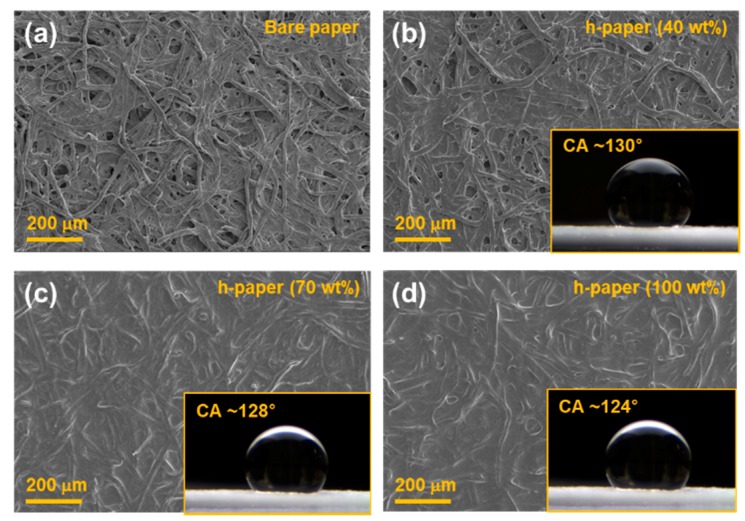
SEM images of the filter paper coated with dilute PDMS of (**a**) 0 wt%, (**b**) 40 wt%, (**c**) 70 wt%, and (**d**) 100 wt% concentrations. The insets show the optical microscopic images of water droplets for contact angle measurements.

**Figure 3 sensors-19-05471-f003:**
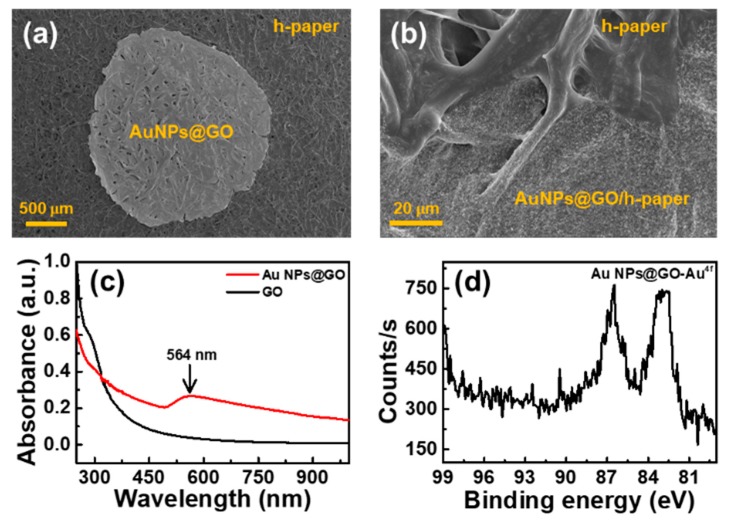
SEM images of the h-paper-based SERS substrate. (**a**,**b**) 50 µL of AuNPs@GO solution was drop-casted and dried on the h-paper substrate. The area of the AuNPs@GO spot was about 4.2 mm^2^. (**c**) UV-Vis spectra of GO and AuNPs@GO. For the AuNPs@GO dispersion (red), a broad peak appeared at around 564 nm, unlike the GO dispersion (black), owing to the localized surface plasmon resonances of AuNPs synthesized on the surfaces of GO. (**d**) X-ray photoelectron spectroscopy (XPS) spectrum of the Au4f peak recorded for AuNPs@GO, indicating that AuNPs were successfully arranged on the surfaces of GO.

**Figure 4 sensors-19-05471-f004:**
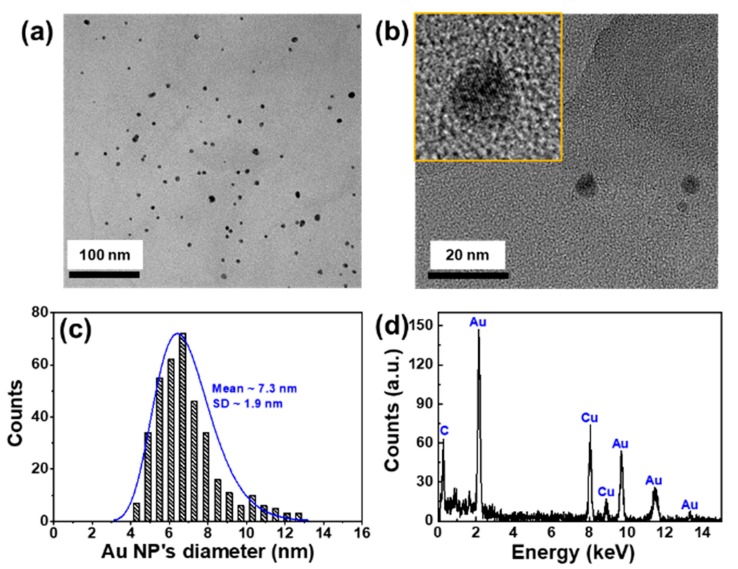
Morphological and chemical compositions of AuNPs@GO. (**a**,**b**) TEM images of the AuNPs@GO at different magnifications. (**c**) Size distribution of the AuNPs on the surface of GO. The mean diameter was about 7.3 nm with a standard deviation of 1.9 nm. (**d**) Energy dispersive spectroscopy of the AuNPs@GO. The characteristic peaks of Au were observed.

**Figure 5 sensors-19-05471-f005:**
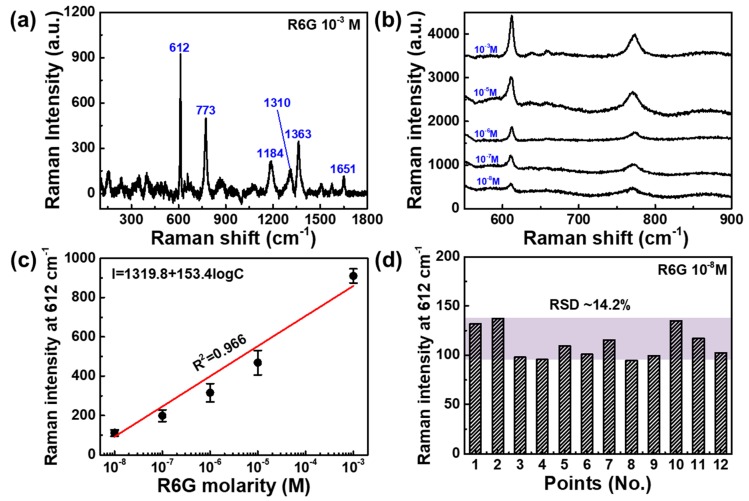
SERS spectra for R6G absorbed on the AuNPs@GO/h-paper. (**a**) R6G spectra with a concentration of 10^−3^ M exhibited a strong peak of the vibrational bands at about 612, 773, 1134, 1310, 1363, and 1651 cm^−1^. (**b**) SERS spectra of R6G molecules with concentrations in the range from 10^−3^ to 10^−8^ M. (**c**) Raman intensity at 612 cm^−1^ as a function of the logarithmic concentration of the R6G molecule. (**d**) Raman intensity at 612 cm^−1^ from 12 randomly selected spots. The relative standard deviation was about 14.2%.

**Figure 6 sensors-19-05471-f006:**
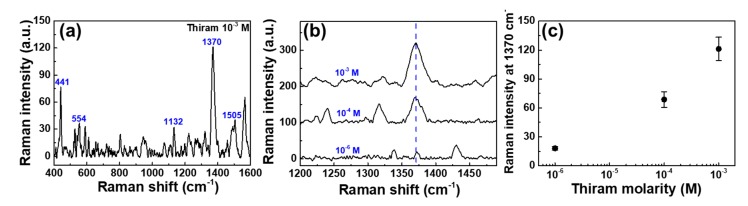
SERS spectra for thiram absorbed on the AuNPs@GO/h-paper. (**a**) Raman spectrum of thiram molecules with concentration of 10^−3^ M exhibited a strong peak of the vibrational bands at about 441, 554, 1132, 1370, and 1505 cm^−1^. (**b**) SERS spectra of thiram molecules with concentrations in the range from 10^−3^ to 10^−6^ M. (**c**) Raman intensity at 1370 cm^−1^ as a function of logarithmic concentration of thiram molecules.

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
