# Peer review of "Hydrophobic Paper-Based SERS Sensor Using Gold Nanoparticles Arranged on Graphene Oxide Flakes"

_sensors, 2019, doi:10.3390/s19245471_

Round 1

Reviewer 1 Report

The article contain a good report about synthesis of novel SERS sensor based on paper with graphene and gold nanoparticles, but still need a major revision probably involving some additional experiments due to reasons mentioned below:

The only test analyte was R6G, it is molecule with some specific properties in water that provide good affinity to hydrophobic surface of sensor. So, testing sensor with a R6G only seems not to be a proper measure of the sensor performance with the typical analytes. There is no analysis of SERS signal of sensor itself, containing a lot of Raman lines from sensor materials. It is good that R6G has two distinct lines shifted from intrinsic ones belonging to the sensor, but other analytes can be not so easy to distinguish. Characterization and measurements part contains a lot of lines containing "model, company, country", so this part should be filled. I also see only one reference in corresponding section that involves gold nanoparticles on carbon structures type of sensors, however, even with colleagues from our institution I can easily provide several more where they are trying to do the same things. I'm not also sure about fit of lin-log graph into the linear model, the fit looks not so good for me, and the low concentration points seems to be off-the-trend, but without some raw data presented I'm not completely sure in this. English language and style of the paper should be improved in several places.

Reviewer 2 Report

--In the abstract, Can you specify what diagnostic applications you are referring to ?, and also about spacecraft? (We expect that our proposed AuNPs @ GO / h-paper-based SERS sensor could be applied to point-of-care diagnostics applications in daily life and in spacecraft.).

--How did you process the Raman spectra, did you make any baseline corrections?

--How did the Raman spectra normalize, in order to take the 612 cm-1 shift as a calibration point?

Reviewer 3 Report

The paper presents an original development of an efficient approach to generate SERS signals. The scientific level of the study makes possible its recommendation for publication. Only some minor revisions are reasonable:

As well as the manuscript is limited by characterization of a functionalized paper in model systems and does not present assays of any analyte in real samples, the comments about point-of care diagnostics in daily life and in spacecraft are premature and should be excluded from the abstract.

The first paragraph of the introduction is overloaded by various examples of papers application and statement of its commonly known properties. It would be reasonable to reduce this historical and educational excursus and to limit references by a few reviews

The authors are right in the statement that theoretical EF of SERS is up to 10 (11) – 10 (12) (lines 43-44). However, the majorities of real practical developments demonstrate much lower EF and decrease sensitivity of the assays at only 2-3 orders. This knowledge should be presented in the introduction to avoid illusions.

The authors' criticism to the earlier described paper SERS sensors (lines 48-53) should be confirmed by some quantitative data (examples), and overcoming the existing limitations should be clear from the Conclusion. Actually the conclusion repeats the Abstract and do not contain any evaluation of the proposed material.

The reached sensitivity of R6G detection should be compared with other developments of materials for the SERS assay with the use of R6G, being a very popular Raman label.

Reviewer 4 Report

The manuscript described a method to fabricate a hydrophobic paper based SERS sensor (h-paper SERS) by hydrophobic modification of the filter paper using PDMS and then coating [email protected] flakes on the hydrophobic paper. In general, the manuscript was well written and it covered a very important application regarding SERS. However, some improvements must be performed before published.

1. When the authors showed their results, didn't demonstrate significant advancements as compared to the previous work in surface modification techniques of paper for SERS, made readers difficulty judging the superiority of the described method. Therefore, it is necessary to provide more analysis.   

2. In this work, the [email protected] solution was stabilized on the h-paper substrate by drop-casting. How to ensure the uniformity of the material on the substrate? The coffee ring effecting could be caused the uneven dispersion of [email protected] I think it will affect the uniformity of the proposed [email protected]/h-paper SERS sensor. Please investigate and explain it.

3.Additional experimental details should be provided for the fabrication of the h-paper-based SERS sensor. The curing time and temperature of PDMS were required. How to remove the solvent.

4. In general, the contact angle decrease as the concentration of the hydrophobic material increases. In this work, why does the contact angle of the modified paper decrease with the increase of the concentration of PDMS, the author needs to discuss.

5. Regarding the measurement of the SERS signal of the [email protected] substrate with R6G molecules, information about reproducibility and reliability are missed.

6. For the detection of R6G, the authors should provide further discussion about the limits of detection that would be required for point-of-care diagnostics applications. Is their assay sufficiently sensitive for use point-of-care diagnostics applications?     

7. More related useful publications can be referenced in this manuscript for improving, for examples

SERS-Based Immunoassays for the Detection of Botulinum Toxins A and B Using Magnetic Beads. By: Kim, Kihyun; Choi, Namhyun; Jeon, Jun Ho; et al. Sensors (Basel, Switzerland) Volume: 19   Issue: 19     Published: 2019 Sep 21

A facile low-cost paper-based SERS substrate for label-free molecular detection, By: Vo Thi Nhat Linh; Moon, Jungil; Mun, ChaeWon; et al. SENSORS AND ACTUATORS B-CHEMICAL Volume: 291   Pages: 369-377   Published: JUL 15 2019

Graphene Oxide Wrapped SERS Tags: Multifunctional Platforms toward Optical Labeling, Photothermal Ablation of Bacteria, and the Monitoring of Killing Effect, By: Lin, Donghai; et al. ACS APPLIED MATERIALS & INTERFACES Volume: 6   Issue: 2   Pages: 1320-1329   Published: JAN 22 2014

A graphene oxide/gold nanoparticle-based amplification method for SERS immunoassay of cardiac troponin I, By: Fu, Xiuli; et al. ANALYST Volume: 144   Issue: 5   Pages: 1582-1589   Published: MAR 7 2019

Round 2

Reviewer 1 Report

The major issues with the publication was solved. Some english language style improvement still have to be done. Also one of a new paragraphs has no text formatting (no upper indexes/lower indexes et cetera). So I suggest to accept this publication after minor revision fixing this issues and I suppose no another round of review process is necessary.

Reviewer 4 Report

The authors have adequately addressed my comments, and I feel the other reviewers' comments have been appropriately addressed by revisions and comments as well. Overall, I feel the manuscript's quality has greatly improved and the manuscript should be accepted for publication in its current form.

Author Response

Reviewer: 4

Recommendation: Minor revision

Comments:

The authors have adequately addressed my comments, and I feel the other reviewers' comments have been appropriately addressed by revisions and comments as well. Overall, I feel the manuscript's quality has greatly improved and the manuscript should be accepted for publication in its current form.

Author response: We gratefully appreciate the reviewer’s kind words.